# Analysis of SI-Related *BoGAPDH* Family Genes and Response of *BoGAPC* to SI Signal in *Brassica oleracea* L.

**DOI:** 10.3390/genes12111719

**Published:** 2021-10-28

**Authors:** Qinqin Xie, Hecui Zhang, Dengke Hu, Qianying Liu, Tonghong Zuo, Yizhong Zhang, Yimei Liu, Siru Zhou, Liquan Zhu

**Affiliations:** College of Agronomy and Biotechnology, Southwest University, Chongqing 400715, China; xieqq1995@163.com (Q.X.); zhanghecui@163.com (H.Z.); hudengke1215@163.com (D.H.); lqy1352345868@163.com (Q.L.); zuotongh@163.com (T.Z.); z8300300@163.com (Y.Z.); liuyimei0710@163.com (Y.L.); zhousiru0325@163.com (S.Z.)

**Keywords:** *B. oleracea* L., *GAPDH*, self-incompatibility, family expansion, *GAPC*, cloning and expression analysis

## Abstract

Glyceraldehyde 3-phosphate dehydrogenase (GAPDH) is not only involved in carbohydrate metabolism, but also plays an important role in stress resistance. However, it has not been reported in *Brassica oleracea*. In this study, we performed a genome-wide identification of *BoGAPDH* in *B. oleracea* and performed cloning and expression analysis of one of the differentially expressed genes, *BoGAPC*. A total of 16 members of the *BoGAPDH* family were identified in *B. oleracea*, which were conserved, distributed unevenly on chromosomes and had tandem repeat genes. Most of the genes were down-regulated during self-pollination, and the highest expression was found in stigmas and sepals. Different transcriptome data showed that *BoGAPDH* genes were differentially expressed under stress, which was consistent with the results of qRT-PCR. We cloned and analyzed the differentially expressed gene *BoGAPC* and found that it was in the down-regulated mode 1 h after self-pollination, and the expression was the highest in the stigma, which was consistent with the result of GUS staining. The promoter region of the gene not only has stress response elements and plant hormone response elements, but also has a variety of specific elements for regulating floral organ development. Subcellular localization indicates that the BoGAPC protein is located in the cytoplasm and belongs to the active protein in the cytoplasm. The results of prokaryotic expression showed that the size of the BoGAPC protein was about 37 kDa, which was consistent with the expected results, indicating that the protein was induced in prokaryotic cells. The results of yeast two-hybrid and GST pull-down showed that the SRK kinase domain interacted with the BoGAPC protein. The above results suggest that the *BoGAPDH* family of *B. oleracea* plays an important role in the process of plant stress resistance, and the *BoGAPC* gene may be involved in the process of self-incompatibility in *B. oleracea*, which may respond to SI by encoding proteins directly interacting with SRK.

## 1. Introduction

Self-incompatibility (SI) is a complex genetic mechanism formed in the long-term evolution of plants to prevent self-incompatibility and promote heterosis. Its molecular mechanism is mainly focused on the synergistic effect of SI signal transduction elements on self-pollen germination or pollen tube growth. It is based on the process of phosphorylation activation of SI signal elements and ubiquitin degradation of pollen affinity factors [1].

Glyceraldehyde 3-phosphate dehydrogenase (GAPDH) is ubiquitous in plant plastids (or chloroplasts). GAPDH in chloroplasts is mainly composed of GapA and GapB subunits, which encode genes called *GAPB* and *GAPA*, with NAPD (H) as the coenzyme. Genes encoding GapC subunits in cytoplasm are called *GAPC*, with NAD^+^ as the coenzyme [2]. In addition to participating in the important function of energy metabolism, GAPDH also plays a key role in plant resistance to stress. During the growth of poplar, GAPDH responds to the signal of water shortage [3]. When *Arabidopsis thaliana* transferred from normal growth conditions to heat shock, anaerobes or increased sucrose supply, the level of *GAPC* transcription increased [4]. AtGAPC can interact with phospholipase Dδ (PLDδ) to transduce H_2_O_2_ signals in plant response to drought stress and promote seed oil accumulation [5,6,7]. Rice increased salt tolerance through overexpression of the *GAPC* gene [8]. The *GAPDH* gene interacts with osmotic pressure-activated protein kinase to produce immune response in tobacco after salt treatment [9]. GAPC3 and GAPC4 are induced by anaerobic stress in *Maize* [10]. The transcription factor *TaWRKY40* of wheat positively regulated the expression of *TaGAPC1* to enhance drought resistance, and wheat *GAPDH12* was significantly upregulated in buds under cold, high temperature, drought and salt stress to participate in the response to adversity stress [11,12]. In addition, GAPDH plays an active role in membrane fusion, vascular bundle formation, phosphotransferase activity, nuclear RNA output, telomere binding, DNA replication and repair, viral pathogenicity and other functions [2]. Meanwhile, it has been found in recent years that GAPDH also plays an important role in embryo development, pollen development, root growth, ABA signal transduction and other aspects [13,14].

The research focus of self-incompatibility has shifted from the key factors in self-incompatibility reactions, such as traditional S-locus receptor kinase (SRK), S-locus glycoprotein (SLG) and S-locus cysteine-rich protein (SCR), to a new direction [15]. In the process of the self-incompatibility physiological reaction, the self-pollen falls to the stigma, and the stigma produces anti-adverse rejection, which causes the self-pollen to fail to hydrate or germinate, triggering the programmed cell death (PCD) of the self-flowering pollen. The effect of pollen inducing SI on stigma is similar to that of an external stress stimulator; reactive oxygen species (ROS) are thought to induce different types of PCD during development. It was found that pollen from different species or incompatibility accumulated a high level of H_2_O_2_ when scattered on the stigma, but this level decreased significantly on the compatible stigma [16,17,18]. GAPC not only plays a role in energy supply, but also transmits the ROS signal in PLDδ interaction. Some scholars have pointed out that PLDδ participates in SI as a target of armadillo-repeat containing protein 1 (ARC1) in SI. However, it is not clear whether there is another way to participate in SI, such as the interaction with GAPDH to transfer the high accumulation of H_2_O_2_ in SI reaction. A total of 16 members were identified in the *B. oleracea* genome, and their evolution analysis, structure analysis, promoter analysis, collinearity analysis and expression pattern analysis were carried out. The differentially expressed *BoGAPC* genes in SI transcriptome were further explored, such as cloning, bioinformatics analysis, tissue specificity analysis, promoter activity analysis, subcellular localization, prokaryotic expression and yeast two-hybrid. The purpose of this study is to prove that the protein encoded by this gene is involved in the self-incompatibility reaction of *B. oleracea* and to provide a basis for the further study of *B. oleracea* SI.

## 2. Materials and Methods

### 2.1. Conserved Domain Analysis of GAPDH Family Genes in B. oleracea and A. thaliana

To identify and classify the *GAPDH* gene family members, we searched the *B. oleracea* genome database (http://brassicadb.org/brad/geneFamily.php (accessed on 5 March 2020)), the TAIR genome database (https://www.Arabidopsis.org/index.jsp (accessed on 5 March 2020)) and the Ensembl database (http://plants.ensembl.org/Brassica_Oleracea/Info/index (accessed on 5 March 2020)) to query and download the *B. oleracea* and *Arabidopsis* whole genome sequences, resulting in *B. oleracea* and *Arabidopsis* chromosome sequence information and related annotation files.

Through sequence alignment and conservative region analysis, we found that *B. oleracea* and *Arabidopsis GAPDH* family genes contained Gp_dh_N superfamily or Gp_dh_C superfamily conserved regions. The conservative domains Gp_dh_N (ID: PF00044.24) and Gp_dh_C (ID: PF02800.20) hidden Markov model (HMM) matrix files were searched and downloaded online through the Pfam database (http://pfam.xfam.org/ (accessed on 5 March 2020)). The hmmsearch program was used to search for the genes containing conserved domains in all protein files of *B. oleracea* and *A. thaliana*. The genes were screened based on the e-value value of 1 × 10^−20^, and the screened genes were reconstructed by the hmmbuild program to reconstruct the HMM of the conserved domain of GAPDH protein specific to *B. oleracea* species. The newly constructed HMM was used to screen *GAPDH* family genes in *B. oleracea* and *Arabidopsis* protein databases, and the protein sequence information of *GAPDH* family in *B. oleracea* and *A. thaliana* was obtained.

The conserved motifs of *B. oleracea GAPDH* gene family were analyzed online by MEME (http://meme-suite.org/tools/meme/ (accessed on 9 May 2020)). The search domain number was 10, the minimum amino acid number of motif was 6 and the longest amino acid number was 50. The search results were processed into vector graphics by TBools software (20200512, Guangzhou, China).

### 2.2. Evolutionary Analysis of Members of GAPDH Gene Family in B. oleracea and Related Species

The *GAPDH* gene family was sequentially screened in turnip, *Brassica napus*, maize, rice and tomato according to the method described in Section 2.1. The *GAPDH* gene family members of the seven species obtained were aligned by MUSCLE. The phylogenetic tree of the GAPDH family of *B. oleracea*, *A. thaliana*, turnip, *Brassica napus*, maize, rice and tomato species was constructed by using the maximum likelihood method (ML) with bootstrap set to 1000 by MEGA X software (MEGA X, Philadelphia, PA, USA).

The ratio of the nonsynonymous substitution rate (Ka) and the synonymous substitution rate (Ks) of the GAPDH gene family in B. oleracea was calculated by KAKS software (KaKs_Calculator2.0, Beijing, China) using the CDS sequence files of B. oleracea and Arabidopsis genomes. The KAKS tandem repeat pairs were screened based on similarity of more than 80% and the length of the alignment of the two sequences being more than 75% of the longer sequence. Circos maps of *GAPDH* gene collinearity within *B. oleracea* genomes and between *B. oleracea* and *Arabidopsis* genomes were drawn and analyzed by Circos software (Circos, Vancouver, BC, Canada).

### 2.3. Chromosome Mapping of GAPDH Family Genes in B. oleracea

The location information of *GAPDH* family genes on 9 chromosomes of *B. oleracea* was recorded by using the genome annotation file, and the chromosomal location map of genes was plotted by using Map Gene (http://mg2c.IASk.in/mg2C_v2.1/ (accessed on 13 May 2020)), an online mapping tool.

### 2.4. Transcript Structure of GAPDH Family Genes in B. oleracea and Analysis of Cis-Acting Regulatory Elements in Promoters

We used the *B. oleracea* genome annotation file to draw the transcript structure of the *B. oleracea GAPDH* gene family through the online tool Gene Structure Display Server 2.0 (http://gsds.cbi.pku.edu.cn/ (accessed on 16 May 2020)).

We used Brassica genome annotation files to extract the *B. oleracea* family of *GAPDH* gene translation initiation site about 1500 bp sequence upstream, using the New PLACE online website (https://www.dna.affrc.go.jp/PLACE/?Action=newplace (accessed on 16 May 2020)) to analyze and identify cis-regulating elements in the promoter subregion. The analysis results were extracted and submitted to the online website Gene Structure Display Server 2.0 (http://gsds.cbi.pku.edu.cn/ (accessed on 9 May 2020)) to draw homeopathic components.

### 2.5. Expression Analysis of BoGAPDH Gene Family

According to the obtained *BoGAPDH* family gene sequence, qRT-PCR-specific primers were designed (Appendix A). Using *B. oleracea* tissue, unpollinated stigma, self-pollination for 15, 30 and 60 min and cross-pollination for 15, 30 and 60 min stigma cDNA as templates, the expression pattern was detected by RT-PCR. RT-PCR experiments were performed in separate tubes. (For details of RNA extraction and RT-PCR methods, please refer to the Appendix A.) Using *B. oleracea BoActin* as the internal reference gene, the relative expression level of the target gene was calculated by the 2^−ΔΔCT^ method. The software we used to perform and analyze the RT-PCR results was Excel 2016 and the equipment used was the BIO-Rad CFX Connect Real-Time PCR System.

*BoGAPDH* family genes were scanned and compared with the data of *B. oleracea* SI transcriptome and Xanthomonas campestris pv stress transcriptome (NCBI accession number GSE107720) in our laboratory to screen the differentially expressed *BoGAPDH* genes and speculate the functional differences of differentially expressed genes and family genes.

### 2.6. Experimental Materials

The materials of *B. oleracea* self-incompatibility line A4 and F1 were provided by the Institute of Cruciferae of Southwest University, Chongqing, China, and planted in isolation in the culture base. In late March 2019, A4 materials with consistent growth were selected to be artificially castrated on the day of flowering, and mature A4 and F1 pollens were used for self-pollination (SI) and cross-pollination (CP). The pollination period was 0, 15, 30 and 60 min. After pollination, the pollen on the stigma was quickly swept away and cryopreserved in liquid nitrogen immediately. At the same time, leaves, buds, sepals, petals, anthers and stigmas were stored in liquid nitrogen.

Plant Total RNA Isolation Kit, Plasmid Mini extraction kit, synthesis and sequencing of various primers were performed by Sangon Biotech (Shanghai, China) Co., Ltd.; the reverse transcription kit was purchased from TransGen Biotech (Beijing, China) Co., Ltd.; and agrobacterium tumefaciens GV3101, yeast Y2HGold and prokaryotic expression strain *E. coli* BL21 (DE3) were purchased from Shanghai Weidi Biotechnology Co., Ltd. (Shanghai, China).

### 2.7. Cloning of Target Genes

The corresponding *B. oleracea* sequences were obtained by comparing the NCBI database and the *B. oleracea* database. According to the principle of homologous recombination and the characteristics of yeast expression vector, primers BoGAPC-AD-F and BoGAPC-AD-R (Appendix A) were designed with Primer 6.0 software (Primer 6.0, Vancouver, Canada). Using *B. oleracea* stigma gDNA and cDNA as templates, the concentration of gDNA and cDNA used in the PCR reactions was 500 ng µL^−1^, the volume used was 1 µL and the amount of gDNA and cDNA used as templates in the PCR was 500 ng. The coding sequence of BoGAPC was amplified by PCR in PrimerSTAR Max DNA Polymerase (TaKaRa, Beijing, China) 25 µL reaction system. The reaction system was as follows: 1 µL of DNA template, 1 µL of BoGAPC-AD-F and BoGAPC-AD-R, 9.5 µL of ddH_2_O and 12.5 µL of 2 × PCR Mixster Mix. The reaction program was as follows: 94 °C pre-denaturation 3 min; 94 °C 30 s, 59 °C 30 s, 72 °C extension 1 min 10 s, 35 cycles; and 72 °C final extension 5 min. The amplified products were sent to Sangon Biotech (Shanghai) Co., Ltd. for sequencing. After correct sequencing, the target fragment was ligated with pGADT7 vector and transferred into *E. coli* competent cell DH5α. The positive clones were selected by PCR and sent to Sangon Biotech (Shanghai) Co., Ltd. for sequencing. According to the above method, the primers SRKj-AD-F and SRKj-AD-R were designed to amplify the kinase domain of the self-incompatibility intermediate protein SRK (Appendix A), and ligated to yeast expression vector pGBKT7.

### 2.8. Bioinformatics Analysis

The bioinformatics analysis of the target gene was carried out by using either online websites or software. The amino acids of the coding sequence of the target gene were deduced by Bio-soft (http://www.biosoft.com/w/overview.html (accessed on 9 May 2020)) and DNAMAN 8.0 (https://www.lynnon.com/ (accessed on 9 May 2020)). The physical and chemical properties of the protein encoded by the target gene were analyzed by the online website ExPASy-ProtParam tool (http://www.expasy.org/ (accessed on 9 May 2020)), the transmembrane domain of the protein was predicted by Signalp (http://www.cbs.dtu.dk/services/SignalP/ (accessed on 9 May 2020)), the transmembrane domain of the protein was predicted by TMHMM (http://www.cbs.dtu.dk/services/TMHMM-2.0/ (accessed on 9 May 2020)) and the phosphorylation sites and N-glycosylation sites were predicted by Netphos (http://www.cbs.dtu.dk/services/NetPhos/ (accessed on 9 May 2020)) and NetNGlyc (http://www.cbs.dtu.dk/services/NetGlyc/ (accessed on 9 May 2020)), respectively. ProtScale (https://web.expasy.org/protscale/ (accessed on 12 May 2020)) predicted the hydrophobicity/hydrophilicity of the target protein, SMART (http://smart.embl-heidelberg.de/ (accessed on 12 May 2020)) was used to analyze the advanced domain of the protein, PROSITE (http://prosite.expasy.org/ (accessed on 12 May 2020)) was used to analyze the functional sites of the protein, the cis-acting elements of the promoter of the target gene were analyzed by PlantCARE (http://bioinformatics.psb.ugentbe/webtools/plantcare/ (accessed on 12 May 2020)) and the software MEGA X and the NCBI (http://www.ncbi.nlm.nih.gov/ (accessed on 26 May 2020)) database were used to construct the evolutionary tree.

### 2.9. Analysis of BoGAPC Expression

The expression of *BoGAPC* was analyzed using the experimental method described in Section 2.5.

### 2.10. Promoter Activity Analysis

The sequence of about 1500 bp upstream of the promoter of the target gene was found on NCBI as the target fragment. Using primer BoGAPC-GUSF/BoGAPC-GUSR (Appendix A), the target fragment was amplified with *B. oleracea* stigma gDNA as a template, and the fusion expression vector was constructed with a pCAMBIA 1391 binary expression vector. The plasmid was extracted and transformed into Agrobacterium tumefaciens GV3101. After detecting the positive transformants, the positive transformants were cultured according to 1:100. The wild type *A. thaliana* was transformed into wild type *A. thaliana* by inflorescence infection, and the hygromycin resistance was screened. The homozygous transgenic *A. thaliana* was sown by screening, and the flowers, buds, leaves and pods were placed in GUS staining solution and incubated overnight at 37 °C. The flowers, buds, leaves and pods of wild type *A. thaliana* were stained and decolorized with 75% alcohol, observed and photographed under stereomicroscope.

### 2.11. Subcellular Localization

According to the coding sequence of the target gene, the green fluorescent gene GFP was ligated according to the homologous recombination method mentioned above, and the vector used was PAN580. *A. thaliana* protoplasts were prepared from *A. thaliana* growing about 4 weeks old and flat leaves with an orientation angle of about 45°. The recombinant plasmids and empty plasmids with a concentration of about 1000 ng/μL were transferred into *Arabidopsis* protoplasts by the PEG method and cultured at 23 °C for 12–16 h. The localization of the target protein was observed under a laser confocal microscope.

### 2.12. Prokaryotic Expression

The target gene was ligated with the prokaryotic expression vector pGEX-4T-1. After the correct sequencing, the strain was expanded and cultured, and the plasmid was extracted. The next steps were to transform the plasmid into *E. coli* BL21 (DE3), perform PCR detection, expand the positive transformant according to 1:100 and inoculate 100 μL of the bacterial solution in 100 mL of 100 μg mL^−1^ ampicillin-resistant LB liquid medium. Then, we placed the culture on a shaker at 37 °C at 225 r/min until the OD600 reached about 0.5–0.6, and added IPTG to a final concentration of 1 mmol L^−1^. After overnight induction at 16 °C, the bacteria were collected for ultrasonic fragmentation, the supernatant was collected by centrifugation, the target protein was purified by beaver GST fusion protein and the expression of the target gene protein was detected by SDS-PAGE electrophoresis. The uninduced target gene protein, induced unpurified target gene protein, induced purified target gene protein and induced purified pGEX-4T-1 empty protein were compared.

### 2.13. Yeast Two-Hybrid Interaction Identification and Protein Interaction Detection by GST Pull-Down

The next steps were to connect the target gene and the BoSRK kinase region to the yeast expression vector pGADT7 and pGBKT7, respectively. When the sequencing was correct, the two recombinant plasmids were extracted and transformed into yeast-competent cells by the polyethylene glycol/lithium acetate method. A negative control (PGADT7-T × PGBDT7 Lam), a positive control (PGAGT7-T × PGBDT7-53) and the experimental group (pGADT7-BoGAPC × pGBKT7-BoSRKj) were competent and plated onto auxotrophic SD/-Trp/-Leu solid media and incubated at 30 °C, inverted for 4 to 5 days to observe plaque growth. The single colony pick point of the growing plaque was smeared on the nutrient-deficient QDO (SD/-Leu/-Trp/-His/Ade) solid medium and cultured at 30 °C upside down for 4–5 days to observe whether there was colony growth.

BoSRKj was ligated into the pET15b expression vector, and the extracted plasmid was transferred into *E. coli* (DE3) to induce expression. The BoGAPC-GST protein purified by 2.12 GST magnetic beads was incubated with BoSRKj-His overnight at 4 °C for protein binding assay, and the obtained protein was detected by electrophoresis.

## 3. Results

### 3.1. Evolutionary Analysis of GAPDH Gene in B. oleracea L.

*GAPDH* family genes contain conserved regions of the Gp_dh_N superfamily or the Gp_dh_C superfamily, which is consistent with the conservative regions of *GAPDH* genes in other plants, indicating that this gene family is relatively conservative in all plant evolution. In the conserved domain of *GAPDH* family genes, the conserved domain of the Gp_dh_N superfamily is the N-terminal NAD^+^ binding domain, and the conserved region of the Gp_dh_C superfamily is the C-terminal catalytic domain [19]. Through the combination of the *B. oleracea* genome database and the Tair genome database, the newly constructed GAPDH conserved domain hidden Markov matrix model was used to screen *GAPDH* family genes [20]. We identified 16 members of the *BoGAPDH* gene family and 12 members of the *AtGAPDH* gene family (Table 1).

In order to better understand and verify the phylogenetic relationship among members of the *GAPDH* gene family in *B. oleracea*, 16 *BoGAPDH*, 12 *AtGAPDH*, 28 *BnGAPDH*, 13 *BrGAPDH*, 8 *OsGAPDH*, 11 *SlGAPDH* and 55 *ZmGAPDH* genes were obtained by using the Maximum Likelihood (ML) method to construct the phylogenetic tree (Figure 1).

The phylogenetic tree can be divided into five groups according to their phylogenetic relationship and evolutionary relationship. Group I is a cruciferous heterosexual group composed of 26 *GAPDH* genes, Group II is a *Gramineae*-specific group composed of 23 *GAPDH* genes, Group III is an eggplant heterosexual group composed of 6 *GAPDH* genes, Group IV is composed of 14 cruciferous genes and 1 *Solanaceae GAPDH* gene and the final group is composed of *Cruciferae*, *Gramineae* and *Solanaceae*. The closest relative of the GAPDH family proteins is between *B. oleracea* and *Arabidopsis*, turnip and *Brassica napus*, the second-most distant relative of tomato and the most distant relative of *Maize* in *Poaceae* rice. The long branching path length may represent the differentiation time successively; most of the *B. napus GAPDH* genes are found on the same branch as those of *A. thaliana*, *B. rapa* and *B. napus*, and the degree of gene sequence similarity between species genomes is higher than that within species, indicating strong gene homology and possibly similar biological functions.

### 3.2. Structure and Analysis of GAPDH Gene Family Members in B. oleracea L.

The protein domains encoded by *B. oleracea GAPDH* gene family members were analyzed online by MEME software. It was found that there were 10 conserved motifs in *B. oleracea GAPDH* family members (Appendix A). The length of these motifs varied from 6 to 50 amino acids. The conserved motif E-value, starting position and length are shown in Appendix A.

Through the conservative motif and structure distribution map, it can be seen that the conserved motifs of *BoGAPDH* genes with similar genetic relationships are basically the same or similar. In terms of the number of conserved genes, most genes have nine conserved motifs, only one gene has four conserved motifs and two genes have three conserved motifs, indicating that *B. oleracea GAPDH* family genes are relatively conservative in the process of evolution. However, the similarity of CDS and intron in number and distribution is lower than that of conservative motifs (Figure 2). In the transcript structure analysis, there are two genes with the same conservative motif, but there are significant differences in the number and location of CDS, indicating that the evolution process of *B. oleracea GAPDH* family genes is more complex. The distance between conserved motifs and relatively conserved motifs of *Bo02686s010.1* gene and *Bo2g075590.1* gene is the same, but there is a significant difference in gene length, CDS and intron distribution, indicating that the evolutionary mechanism is more complex, which may lead to significant differences in function, and it is possible to participate not only in SI signal transduction but also in physiological and biochemical responses to stress.

### 3.3. Chromosome Mapping of GAPDH Gene in B. oleracea

Through the chromosome location of the gene, the distribution of the identified *BoGAPDH* gene on the chromosome was detected, and it was found that the *GAPDH* gene family was not uniformly distributed in the *B. oleracea* genome. As shown in Figure 3, a total of 16 *BoGAPDH* genes are mapped to chromosomes C1, C2, C3, C5, C6, C7 and C8. *Bo00285s370.1* and *Bo02686s010.1* are located on chromosome scaffolds Scaffold00285 and Scaffold02686, respectively. Chromosomes 1, 3, 6 and 7 and Scaffold00285 and Scaffold02686 show single gene distribution; there are two genes on chromosome 2 and four genes are distributed on chromosomes 5 and 8. There are three genes on chromosome 5 and there is a small physical distance between the two genes on chromosome 8, so there may be tandem replication events.

### 3.4. Cis-Acting Regulatory Elements of B. oleracea GAPDH Gene

The promoter is a DNA sequence that binds to transcription factors and RNA polymerase to regulate the downstream genes of the promoter, which plays an extremely important role in gene transcriptional expression. Various response elements on the promoter are the key regions for its response to various environmental factors, such as auxin response, light response, stress response, etc. [21]. The purpose of this study is to elucidate the possible regulatory mechanism of 16 screened *BoGAPDH* family genes in abiotic or biological stress responses. The upstream 1.5 kb sequences of the translation initiation sites of these genes were extracted and submitted to the PLACE database for analysis and identification of cis regulatory elements in the promoter region. The results showed that the main regulatory elements detected in the 16 gene promoters included: ERELEE4: ethylene response element; IBOXCORE: light response element; MYCCONSENSUSAT: response element related to dehydration reaction and abscisic acid signal transduction in *Arabidopsis*; POLLENLELAT52: pollen-specific cis-acting element; WBOXATNPR1: pressure- and defense-related response element; and WRKY71OS: anti-elements related to gibberellin signaling (Figure 4).

The ERELEE4 regulatory element is related to the ripening and senescence of plant seeds and fruits, and may also be involved in the regulation of plant circadian rhythm. This element participates in the growth of tomato and plays an important role in fruit ripening. In addition, it also plays an active role in the senescence mechanism of carnation and stress response of Chilean tomato [22,23]. IBOXCORE regulatory element is a conserved sequence upstream of photoregulatory genes in monocotyledons and dicotyledons, related to photoregulated transcription. Some scholars have pointed out that the mutation of this regulatory element will seriously affect the expression of light-regulated genes in *Arabidopsis*, while the growth and development of plants sense the changes in light in the external environment through the light receptor and activate a series of signal transduction processes, thus flowering in time and completing the fruiting [24,25,26]. Flowering time in *B. oleracea* may affect subsequent pollination or the low quality of pollen may lead to a decrease in seed yield, and the molecular mechanism of this regulatory element mediating light signal to regulate flowering may have a greater impact on flowering and seed setting in *B. oleracea*. The MYCCONSENSUSAT regulatory element is a promoter region recognition site of the dehydration response gene rd22 in *A. thaliana*, and the MYC recognition site has a cis-acting element role in drought-induced *rd22* gene expression. Subsequent studies have shown that the AtMYC2 protein plays a role as a transcriptional activator in ABA-inducible gene expression under drought stress in plants, and MYCCONSENSUSAT has also been reported as a cold response regulatory element in *Arabidopsis* [27,28,29,30,31]. POLLENLELAT52 is a pollen-specific cis-acting element that exists in the promoters of *BoGAPDH* family genes. Studies have demonstrated that this regulatory element is involved in the precise pollen development and tissue-specific expression of LAT52 in tomato, and can affect the transcriptional activation of LAT52 [32,33]. Other scholars have shown that the 5′-upstream region of LeMAN5 endonuclease-β-Mannanase, which is related to tomato pollen germination and pollen tube elongation, contains four copies of pollen-specific cis-acting element POLLEN1LELAT52 (AGAAA), which can be used to control pollen fertility. This physiological regulation of pollen may play an important role in the process of self-incompatibility in *B. oleracea* [33]. The WBOXATNPR1 regulatory element plays an important role in responding to environmental stress and enables plants to acquire disease resistance through pathogen-induced enhancement of the *Arabidopsis* transcription factor *AtWRKY18* [34,35,36]. The WRKY71OS regulatory element is the transcriptional repressor of the gibberellin signal transduction pathway [37]. The above cis-acting element analysis showed that these *B. oleracea GAPDH* genes could respond to different biotic or abiotic stresses and were closely related to the development of *B. oleracea* pollen.

### 3.5. Tandem Repeat Gene Pair and Collinearity Analysis of GAPDH Family Genes in B. oleracea and A. thaliana

The expansion of plant gene family is mainly through gene fragment replication and tandem replication, so the gene copy event on the genome can be used as a reference standard to judge the degree of gene evolution. The ratio of the non-synonymous substitution rate (Ka) to the synonymous substitution rate (Ks) was calculated by KAKS software in the *B. oleracea GAPDH* gene family. The main screening criteria for KAKS tandem repeat gene pairs are the similarity of the latter two sequences being more than 80%, and the length of the alignment of the two sequences being more than 75% of the longer sequence [38].

We obtained 16 pairs of repetitive genes in the *BoGAPDH* family (Appendix A). The KaKs of these 16 pairs was less than 1, indicating that their purification selection reduced the rate of amino acid change. In order to further study the replication of *BoGAPDH* family genes in *B. oleracea*, we downloaded relevant genomic information from the *B. oleracea* and *Arabidopsis* databases to find tandem repetitive gene pairs within and between genomes, and then plotted Circos maps to analyze the collinear relationship. Figure 5A is based on the chromosome location information of *B. oleracea* and *A. thaliana* and 17 pairs of tandem repeat genes selected, and the intergenomic collinearity analysis map was drawn by Circos software. The red lines indicate the collinearity of *GAPDH* family members between genomes. Figure 5B is an intragenomic collinearity analysis map drawn by Circos software based on the location information of nine chromosomes in the *B. oleracea* genome and 14 pairs of tandem repetitive genes screened. The red lines represent the collinearity of *GAPDH* family members in the *B. oleracea* genome and the different colors of the background are the collinear blocks drawn according to the collinear gene pairs in the whole genome of *B. oleracea*.

### 3.6. Expression Analysis of GAPDH Family Genes in B. oleracea

#### 3.6.1. Expression Analysis of *GAPDH* Family Genes in *B. oleracea* during Self-Pollination and Cross-Pollination

In order to further the differential expression of *GAPDH* family genes in *B. oleracea* after self-pollination and cross-pollination, 14 genes were selected for expression analysis by qRT-PCR. As shown in Appendix A, in self-pollination 0–60 min, the expression pattern of *Bo5g017500.1* and *Bo8g107880.1* was first up-regulated and then down-regulated, and all other genes were down-regulated in 0–15 min, which was the critical period of SI reaction. In 30–60 min, the expressions of *Bo7g083800.1*, *Bo1g152180.1*, *Bo6g080970.1*, *Bo02686s010.1* and *Bo5g016460.1* were gradually up-regulated and then down-regulated. *Bo8g065470.1* and *Bo8g104720.1* were first down-regulated and then up-regulated in self-pollination 0–60 min, and generally showed a down-regulation trend, while *Bo00285s370.1*, *Bo2g095350.1*, *Bo8g024290.1* and *Bo5g021670.1* showed a down-regulation trend after self-pollination. In cross-pollination 0–60 min, *Bo5g017500.1* and *Bo8g107880.1* were up-regulated at first and then down-regulated, *Bo7g083800.1* and *Bo02686s010.1* were down-regulated at first and then up-regulated, *Bo5g016460.1* and *Bo8g024290.1* were up-regulated at first and then down-regulated and then up-regulated, but the expression of these six genes showed an up-regulation trend as a whole. The results showed that most of the genes were consistent with the results of transcriptome analysis.

#### 3.6.2. Tissue Expression Analysis of *GAPDH* Gene in *B. oleracea*

The qRT-PCR results of *B. oleracea* tissue showed that most of the genes were highly expressed in stigma and sepals, and the genes highly expressed in stigma may be directly involved in the signal transduction process of stigma SI. In addition, *Bo7g083800.1* was also highly expressed in pollen, *Bo02686s010.1* was highly expressed in petals, *Bo5g017500.1* was highly expressed in stamens and *Bo8g107880.1* was highly expressed in leaves and stems (Appendix A).

#### 3.6.3. Expression Analysis of *BoGAPDH* Gene in *B. oleracea* Transcriptome

There are 16 members of the *BoGAPDH* gene family, and the functions of the coding proteins are also different. This gene family not only plays an important role in glycolysis and stress resistance, but also participates in *B. oleracea* self-incompatibility in some way. To explore whether the genes encoding proteins in this family all have the above functions, or if only some proteins exert one function or multiple functions, we combined *B. oleracea* SI transcriptome data analysis and found that in the *BoGAPDH* family, most genes were significantly differentially expressed 60 min after self- versus cross-pollination (Figure 6A). It may be that most genes in this family are involved in the regulation of the SI pathway in *B. oleracea*.

The transcriptome data related to stress were searched by NCBI, and the transcriptome data of *B. oleracea* stressed by Xanthomonas campestris were selected (NCBI accession number is GSE107720). It was found that there were three genes differentially expressed in the *BoGAPDH* family on the third day of stress (Figure 6B), which were *Bo00285s370.1*, *Bo3g057250.1* and *Bo7g083800.1*. On the 12th day, *Bo7g083800.1* was differentially expressed, and *BoGAPDH* differentially expressed genes were all up-regulated. This shows that *B. oleracea* adapts to stress by up-regulating the coding protein expression of these three genes. *BoGAPDH* family genes play an important role in the process of disease resistance and stress in *B. oleracea*, and *GAPDH* family genes encode proteins with diverse functions, which do not play a single functional role during plant growth, and different genes encode proteins that carry out different physiological functions.

#### 3.6.4. Full-Length Cloning of BoGAPC Gene

The results of *Bo8g065470.1* cloning differentially expressed in SI transcriptome showed that the size of the gDNA PCR product was about 2000 bp; for cDNA PCR, the product was about 1000 bp (Figure 7). The results of sequencing showed that the length of gDNA sequence obtained by PCR amplification was 2061 bp, and cDNA sequence length was 1020 bp. Combined with NCBI alignment, it was found that it was a coding sequence of cytoplasmic glyceraldehyde 3-phosphate dehydrogenase, so the gene was named *BoGAPC*.

#### 3.6.5. Protein Structure Analysis of BoGAPC

The structural characteristics of BoGAPC protein were analyzed by DNAMAN software and ExPASy ProParam online software (Appendix A). The protein consists of 339 amino acids with a relative molecular weight of 37.07 kD, a theoretical isoelectric point of pI of 5.01, a total number of positively charged and negatively charged amino acids of 0 and an instability index of 45.88. It belongs to an unstable protein with a short half-life and a fat index of 26.37. The hydrophobicity index was 0.747, and the prediction of subcellular localization of the protein showed that it was located in the cytoplasm, so the protein may be a hydrophilic protein. The protein has two N-glycosylation sites, but it does not contain any signal peptide or a transmembrane helix region, and it is not a membrane protein or secretory protein, so it will not be glycosylated, although there are potential glycosylation sites. In this protein, 15 serine (Ser), 10 threonine (Thr) and 3 tyrosine (Tyr) may be protein kinase phosphorylation sites. The protein contains a conserved domain of Gp_dh_N, which is located between the amino acid residues of 7157 amino acid. The three-dimensional structure prediction of BoGAPC was obtained by using the online website Expasy (https://swissmodel.expasy.org/ (accessed on 21 May 2020)) (Appendix A).

#### 3.6.6. Protein Evolution Analysis of BoGAPC

Phylogenetic tree construction of BoGAPC with corresponding protein amino acid sequences within other species using Mega x software revealed that BoGAPC is the closest related relative to BnGAPC in oilseed rape and the furthest related to AtGAPC (Appendix A). In order to clarify the homology of GAPC proteins among different species, we compared them with rape BnGAPC, turnip BrGAPC, *A. thaliana* AtGAPC, dicotyledonous Solanaceae tomato SlGAPC, monocotyledonous Gramineae rice OsGAPC and maize ZmGAPC. The results showed that there was a relatively conserved protein fragment between monocotyledonous and dicotyledonous species, that is, the conserved domain of Gp_dh_N (Appendix A).

#### 3.6.7. Expression Analysis of *BoGAPC* Gene

The expression of *BoGAPC* was detected by qRT-PCR at various time periods after different pollination treatments in kale, and the results indicated that the gene was in a down-regulated expression mode from 0 to 1 h after self-pollination, thus suggesting that the apparent down-regulation of the gene after self-pollination may be in response to the signal of self-incompatibility, and the down-regulation of *BoGAPC* expression within 1 h of self-pollination may be the result of the onset of SI response. By combining the subsequent results that the protein encoding this gene may interact with SRK, a key factor of SI response, *BoGAPC* may negatively regulate the occurrence of SI response in kale by interacting with SRK. The expression of this gene was found to be the highest in the stigma, followed by the sepals, and the process of self-incompatibility in kale is that the self-pollen falls on the stigma, and the stigma generates rejection resistance, resulting in the inability of the self-pollen to hydrate or germinate. The expression of this gene is the lowest in the pollen, which produces a tenfold difference from the expression in the stigma. This is consistent with the pattern of self-incompatibility (Figure 8).

#### 3.6.8. Analysis of the Specificity of the Promoter of the *BoGAPC* Gene

Through the analysis of the *BoGAPC* promoter region, several elements were found: abscisic acid response element, gibberellin response element, ethylene response element, light response element, defense response element, flower-specific cis-acting element, carbon metabolism-related element and so on (Appendix A). These findings indicate that this gene, in addition to playing an important role in carbon metabolism, is involved in other physiological processes through a variety of cis-acting elements in addition to hormone response elements and multiple specific acting elements involved in floral organ development. This gene may also be involved in SI through plant hormone response elements as well as pollen-specific acting elements.

In order to see more intuitive results of promoter analysis in plant tissues, we constructed an expression vector with β-glucuronidase (GUS) as a reporter gene promoter. The successfully constructed BoGAPC-pCAMBIA1391 binary expression vector was transformed into *A. thaliana* by inflorescence infection to obtain transgenic plants. After screening to homozygous, leaves, petals, buds and pods were immersed in GUS dye solution and stained at 37 °C. After staining, the plants were decolorized with ethanol, and the wild type *A. thaliana* with the same growth was selected as the control. As shown in Figure 9, *BoGAPC* was highly expressed in pods, sepals and stigmas, higher at the top and base of pods than in shells, lower in leaves and petals and mainly at the edge of leaves (Figure 9).

#### 3.6.9. Subcellular Localization of BoGAPC

By constructing the BoGAPC-PAN fusion expression vector and transferring it into the protoplasts of *A. thaliana*, it was found that BoPGAPC was obviously located in the cytoplasm and belonged to the active protein in the cytoplasm, which was consistent with the characteristics of no signal peptide and transmembrane domain of the protein (Figure 10).

#### 3.6.10. Prokaryotic Expression of BoGAPC

The protein expressed by the recombinant strain linked to BoGAPC and pGEX-4T-1 was purified by beaver GST fusion protein, and the expression of the target protein was detected by SDS-PAGE electrophoresis (Figure 11). From left to right, there are protein markers. Lane 1 is uninduced pGEX-4T-1-BoGAPC protein, lane 2 is induced unpurified pGEX-4T-1-BoGAPC protein, lane 3 is induced purified pGEX-4T-1-BoGAPC protein and lane 4 is induced purified empty protein. The size of the protein expressed in empty pGEX-4T-1 was about 26 kD, and the protein band in lane 3 was about 65 kD. The predicted size of BoGAPC protein was about 37 kD, and the size of the fusion protein was about 63 kD. The size of the target protein was consistent with the predicted results, indicating that the pGEX-4T-1-GAPC fusion protein was successfully induced in prokaryotic cells.

#### 3.6.11. Interaction between BoGAPC and BoSRKj

After successful ligation of BoSRKj and pGBKT7 yeast expression vector, the plasmid was extracted for self-activation detection. Under the premise of no self-activation, yeast two-hybrid was used to determine the interaction between BoGAPC and BoSRK kinase domain. The results of yeast double hybrid showed that the experimental group (pGADT7-BoGAPC × pGBKT7-BoSRKj), the negative control group (pGADT7-T × pGBDT7-Lam) and the positive control group (pGADT7-T × pGBDT7-53) were transformed into yeast-receptive cells and coated on SD/-Trp/-Leu plate. After 3 days of inverted culture at 30 °C, leukoplakia grew in both the experimental group and the positive control group, while the plaque in the negative control group was red (Figure 12A). The plaque of the experimental group was detected by PCR to expand the target band, indicating that the recombinant plasmid was transformed into the yeast cells. The next steps were to dilute the single spot on the second plate with sterile water 100 times and apply it to the SD/-Trp/Leu/-His/Ade plate, and invert it for 3–5 days. We found that both the experimental group and the positive control can be on the plate. The negative control cannot grow normally, indicating that there is an interaction between BoSRKj and BoGAPC.

The interaction between BoGAPC and BoSRKj was verified by in vitro GST pull-down (Figure 12B).

## 4. Discussion

### 4.1. Members of GAPDH Family Participate in Plant Stress Resistance

When *GAPC* gene mutation occurs in *A. thaliana*, there are fertility defects, changes in embryonic development and morphological changes in long-horned fruit. Abortion and empty embryo sacs are shown in the basal and apical pods, respectively, which changes the development of seeds and fruits, and significantly reduces the seed setting rate [39]. When GAPC is inhibited in *Brassica rape*, it may affect tapetum and microspore development, thus causing abortion in rapeseed [40]. In *A. thaliana*, cytoplasmic GAPDH (GAPC) can be ubiquitinated by E3 ubiquitin protein ligase SINAL7 in vitro, changing its catalytic activity, while E3 ubiquitin protein ligase is associated with various biological processes, such as cell cycle control and regulation of multiple developmental pathways [41,42,43,44]. GAPDH also plays the role of multifunctional protein in plant abiotic stress and plant fertility [45]. Recent studies have shown that overexpression of *OsGAPB* in rice under low light stress increases stress tolerance and reduces the effect of low yield caused by low light stress [46]. The expression of *PPN2* gene in yeast is controlled by the strong constitutive promoter of glyceraldehyde 3-phosphate dehydrogenase gene (*PKG1*), resulting in a significant increase in cobalt/zinc ion-stimulated cohesive phosphatase activity, and cells overexpressing *PPN2* are more resistant to peroxides and alkali [47]. *TaGAPC5/6* genes were found to respond positively to drought stress through reactive oxygen species (ROS) scavenging and stomatal movement in wheat. The results of the study demonstrated that a new positive regulatory mechanism of *TaGAPC* could help wheat fine-tune its drought response. In addition, some scholars identified MYB binding sites (AACTAAA/C) in the promoter sequences of *GAPCp2* and *GAPCp3* identified as a key cis-element in response to drought stress [48,49]. In the analysis of the *GAPDH* gene family in sweet orange (umbilical orange), the GO analysis results of genes interacting with *CsGAPDHs* indicated that *CsGAPDHs* might be involved in the response to inorganic substances. Under the condition of phosphorus deficiency, *CsGAPDH3* and *CsGAPDH* 6 were down-regulated and up-regulated, respectively, suggesting that *GAPDHs* play a role in some phosphorus-related processes [50].

Plants adapt to adversity by up- or down-regulating *GAPDH*, and *B. oleracea* undergoes the same process of self-incompatibility in response to adversity generated by stigma rejection of autogamous pollen. The self-incompatibility transcriptome of *B. oleracea* showed that most members of the gene family showed a down-regulated expression trend during self-pollination. In the transcriptome data of *B. oleracea* under Xanthomonas campestris stress, *BoGAPDH* differentially expressed genes were up-regulated to adapt to bacterial stress. Analysis of *BoGAPDH* family expression during self-pollination and cross-pollination showed that most members of this gene family responded to adversity by down-regulating expression during the self-pollination phase, and this result was consistent with the *B. oleracea* transcriptome data.

### 4.2. Replication and Evolution of GAPDH Gene in B. oleracea

*GAPDH* family genes have been identified in many species, including *A. thaliana*, sweet orange (navel orange), rice, cassava, wheat, etc., indicating that the *GAPDH* family mainly plays a role in plant abiotic stress tolerance [11,50,51,52]. At present, there is no research report on *B. oleracea GAPDH* family genes. In this study, 16 *B. oleracea* GAPDH genes and 12 *Arabidopsis GAPDH* genes were identified in *B. oleracea* and *A. thaliana* genome databases. The phylogenetic relationships of *BoGAPDH* genes in *B. oleracea*, *Arabidopsis*, turnip, *Brassica napus*, maize, rice and tomato were systematically analyzed and compared. The analysis of conserved motif and gene structure revealed the evolutionary relationship, gene expansion and functional diversity of the *GAPDH* gene family in *B. oleracea*. Chromosome mapping and collinearity analysis showed that the *GAPDH* family members in *B. oleracea* genome and between *B. oleracea* and *Arabidopsis* genomes reduced the rate of amino acid change during long-term evolution; the genes were relatively conserved, and these genes may be amplified by the same ancestor gene. A large number of cis-acting elements related to pollen specificity, hormone regulation and dehydration were found in the promoter of *BoGAPDH* family genes, which further proved that these genes play an important role in the regulation of plant development and abiotic stress. Plant SI may be related to plant stress resistance in evolution. This study provides a targeted basis for the selection of genes for follow-up study of *B. oleracea* SI response.

### 4.3. BoGAPC May Be Involved in Self-Incompatibility Signal Transduction in B. oleracea

In *A. thaliana*, GAPDH (GAPC) can be ubiquitinated by E3 ubiquitin protein ligase SINAL7 in vitro, changing its catalytic activity, while E3 ubiquitin protein ligase is associated with various biological processes, such as cell cycle control and regulation of multiple developmental pathways [41,42,43,44]. ARC1 is a typical E3 ubiquitin ligase in *B. oleracea* self-incompatibility, which plays an important role in the signal process of self-incompatibility. At the same time, PUB protein and ARC1 belong to E3 ubiquitin ligase, which may lead to self-incompatibility by negative regulation of pollen germination or interaction with SRK during *B. oleracea* self-incompatibility [53,54,55]. E3 ubiquitin ligase plays an important role in *B. oleracea* self-incompatibility, and BoGAPC may be involved in *B. oleracea* self-incompatibility by interacting with E3 ubiquitin ligase or producing other biological stress responses.

In this study, most of the *BoGAPDH* family genes were down-regulated in 0–60 min of self-pollination, and highly expressed in stigma and sepals, which may be directly involved in SI signal transduction in stigma, resulting in down-regulation of expression. *BoGAPDH* is a small gene family; most of the genes in the SI response exhibit the same expression pattern and may be involved in SI through the same mode of action, and the other physiological functions this gene family remain of significant research value. *BoGAPC* was down-regulated within 1 h after self-pollination. Plant tissue fluorescence quantitative analysis showed that the expression of this gene was the highest in stigma, followed by sepals, and the lowest in pollen. The quantitative results of tissue expression were consistent with the results of GUS staining. Through cis element analysis of the promoter region of the gene, it was found that the gene not only had stress response elements and plant hormone response elements, but also had a variety of specific elements regulating the development of floral organs. Subcellular localization showed that the BoGAPC protein was located in the cytoplasm, and the prokaryotic expression results showed that the size of BoGAPC protein was about 37 kD, which was consistent with the expected results, indicating that the protein was induced to be expressed in prokaryotic cells. 

The results of yeast two-hybrid and GST pull-down showed that the SRK kinase domain interacted with BoGAPC protein, which further proved that it could participate in SI through direct interaction with SRK. *B. oleracea* self-incompatibility is genetically controlled by an S-locus gene with polymorphisms, SLG, with SRK being the two S alleles segregating at the S-locus. SRK is a key self-incompatibility factor in *Brassica*, and the traditional theory of self-incompatibility is based on the interaction of SLG–SRK–ARC1 trios that transduces self-incompatibility signaling downstream to cause stigma cells to become deficient in self-pollen and fail to germinate, leading to the occurrence of self-incompatibility reactions. SRK plays a critical role throughout the signaling process by serving as the binding region for the C-terminal arm repeat region of the downstream substrate ARC1, the current international frontiers in SI signaling. The ARC1 protein is not the only downstream signaling element of SRK—there are still many unknown signaling elements involved in the self-incompatibility signaling process [56]. Our experimental results show that BoGAPC protein can interact with the SRK kinase domain, indicating that the protein may participate in SI through direct interaction with SRK and play an important role in SI signal transduction.

According to the above results, it is inferred that *BoGAPC* may be a gene involved in the process of self-incompatibility in *B. oleracea*.

## 5. Conclusions

In this study, we analyzed the evolution, structure, promoters and collinearity of 16 members of *B. oleracea* BoGAPDH protein family, and analyzed the expression pattern of the genes encoded by this family protein after self-incompatibility. The results showed that the family protein was evolutionarily conservative and played an important role in stress resistance. Through the cloning and expression analysis of the gene encoded by BoGAPC protein, we found that BoGAPC may respond to the self-incompatibility signal by interacting with BoSRK.

## Figures and Tables

**Figure 1 genes-12-01719-f001:**
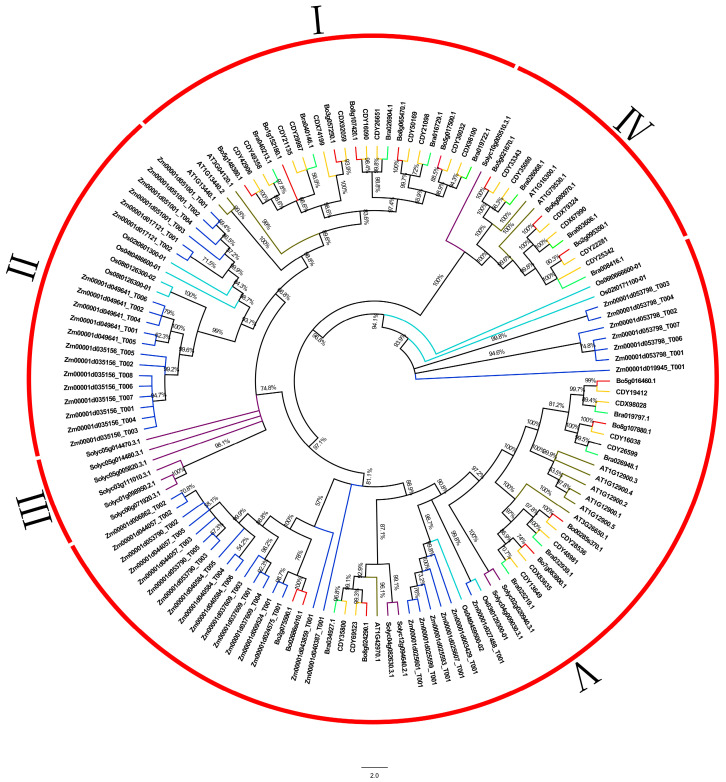
Phylogenetic analysis of GAPDH proteins in *Brassica oleracea*, *Arabidopsis*, turnip, *Brassica napus*, corn, rice and tomato species. The Maximum Likelihood (ML) phylogenetic tree was constructed by using MEGA X software; the bootstrap value is set to 1000. Group I is *Cruciferae*-specific, Group II is *Gramineae*-specific, Group III is *Solanaceae*-specific, Group IV consists of 14 *Cruciferae* and 1 *Solanaceae* GAPDH genes, and the rest, coded as Group V, are *Cruciferae*, *Gramineae* and *Solanaceae*. The red line indicates kale, the dark green line indicates *Arabidopsis*, the light green line indicates turnip, the dark blue line indicates maize, the light blue line indicates rice, the yellow line indicates kale-type oilseed rape and purple indicates tomato.

**Figure 2 genes-12-01719-f002:**
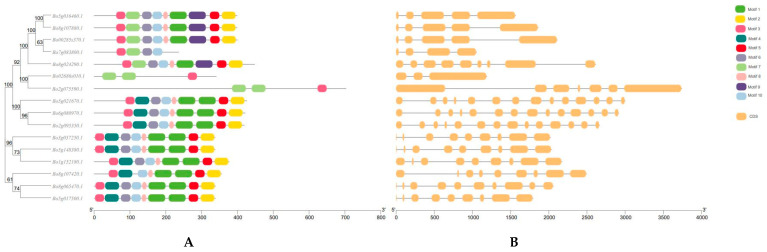
(**A**) Conserved motif distribution of *BoGAPDH* gene; the 10 conservative motifs are represented by different colors. (**B**) The structural distribution of the *BoGAPDH* gene; yellow represents exons, lines represent introns.

**Figure 3 genes-12-01719-f003:**
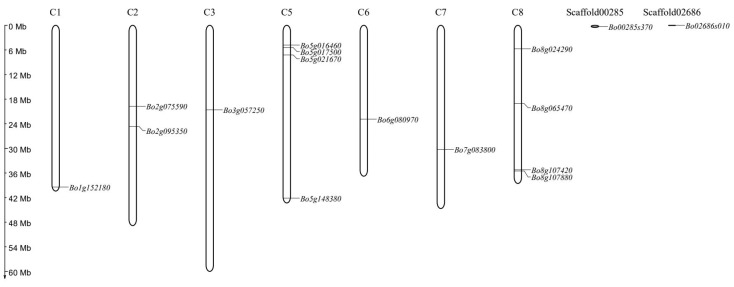
Chromosomal and scaffold distribution of *GAPDH* genes in *B. oleracea*.

**Figure 4 genes-12-01719-f004:**
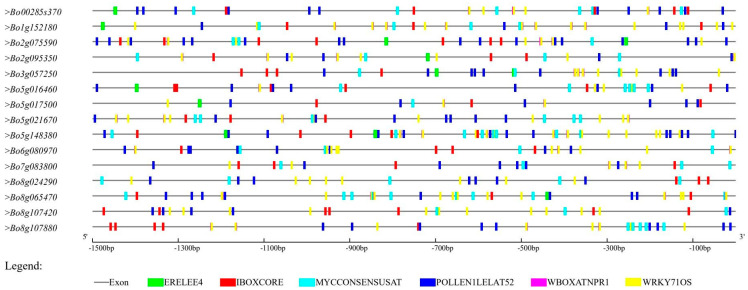
Analysis of cis-acting elements in the promoter region of the *GAPDH* gene in *B. oleracea*. ERELEE4: ethylene response element; IBOXCORE: light response element; MYCCONSENSUSAT: response element related to dehydration reaction and abscisic acid signal transduction in *Arabidopsis*; POLLENLELAT52: pollen-specific cis-acting element; WBOXATNPR1: pressure- and defense-related response element; WRKY71OS: anti-elements related to gibberellin signaling.

**Figure 5 genes-12-01719-f005:**
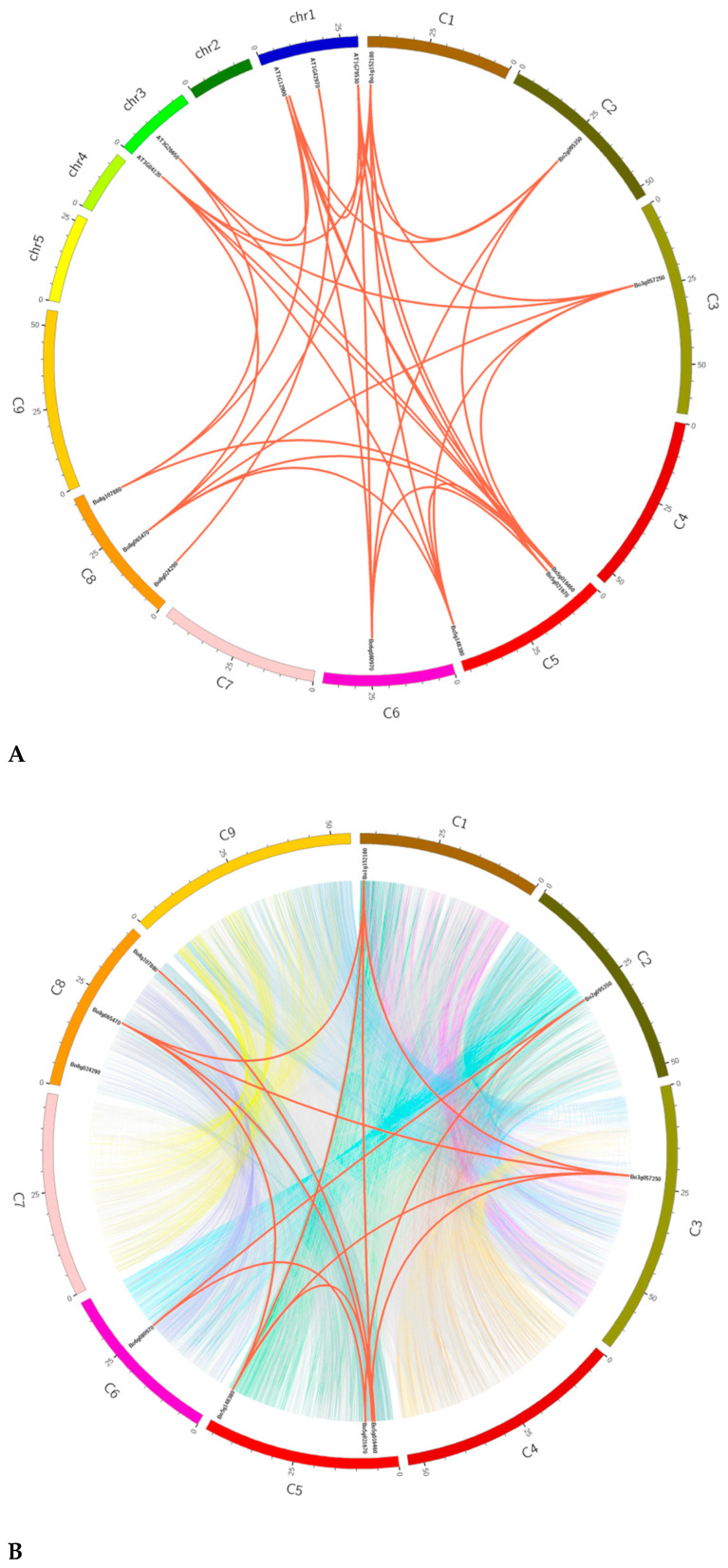
*GAPDH* gene tandem replication in the *B. oleracea* genome and between the *B. oleracea* and *Arabidopsis* genomes. (**A**) *GAPDH* gene tandem replication event between *B. oleracea* C1–C9 and *Arabidopsis* Chr1–Chr5. The solid red line indicates tandem duplication between genes. (**B**) Tandem replication events between the *B. oleracea* C1–C9 chromosomes. The solid red line indicates the replication between *GAPDH* genes, and the blocks with different colors in the background indicate the replication of all genes in the *B. oleracea* genome.

**Figure 6 genes-12-01719-f006:**
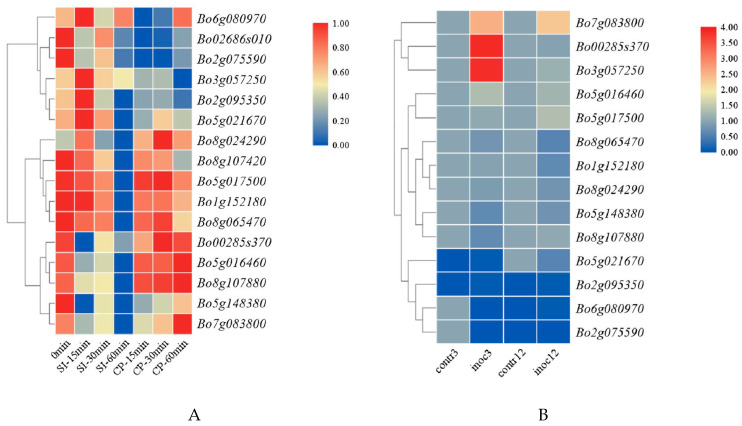
(**A**) Expression patterns of *BoGAPDH* gene family after self-pollination and cross-pollination. (**B**) The expression pattern of *BoGAPDH* gene family stimulated by Xanthomonas campestris pv.

**Figure 7 genes-12-01719-f007:**
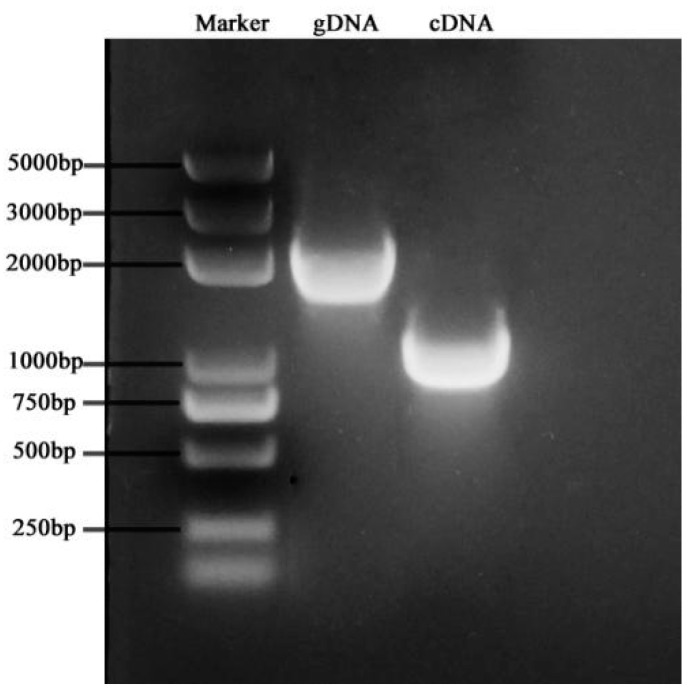
The agarose gel electrophoresis of cDNA and the gDNA and PCR product of BoGAPC.

**Figure 8 genes-12-01719-f008:**
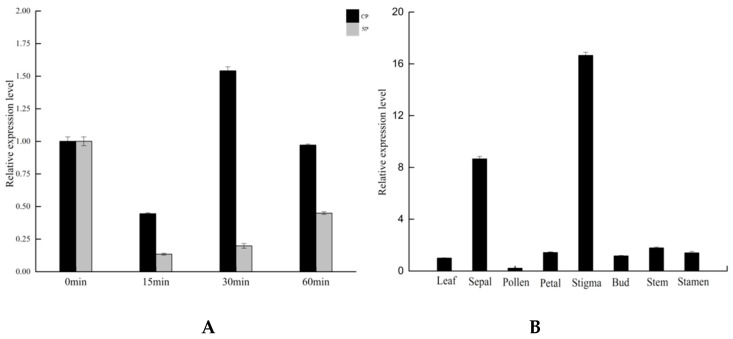
(**A**) *BoGAPC* expression analysis after pollination. The pollination period is 0 min, 15 min, 30 min and 60 min. SP: self-pollination; CP: cross-pollination. (**B**) Tissue expression analysis, *B. oleracea* tissue includes leaves, sepal, pollen, petals, stigma, bud, stem and stamen.

**Figure 9 genes-12-01719-f009:**
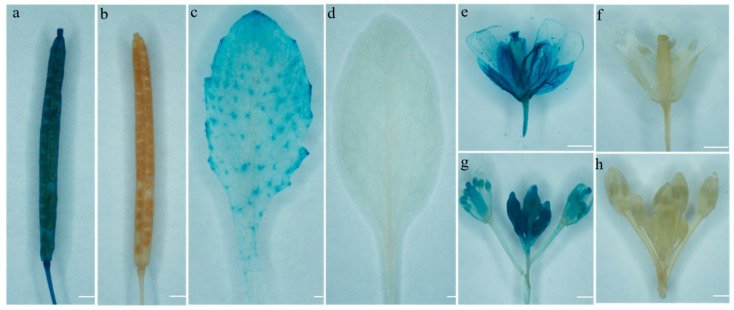
GUS staining diagram. Transgenic *Arabidopsis* was used as the experimental group and Columbia wild type *Arabidopsis* was the control group: (**a**) pods; (**c**) leaves; (**e**) flowers; (**g**) buds; (**b**,**d**,**f**,**h**) CK. Bar = 25 μm.

**Figure 10 genes-12-01719-f010:**
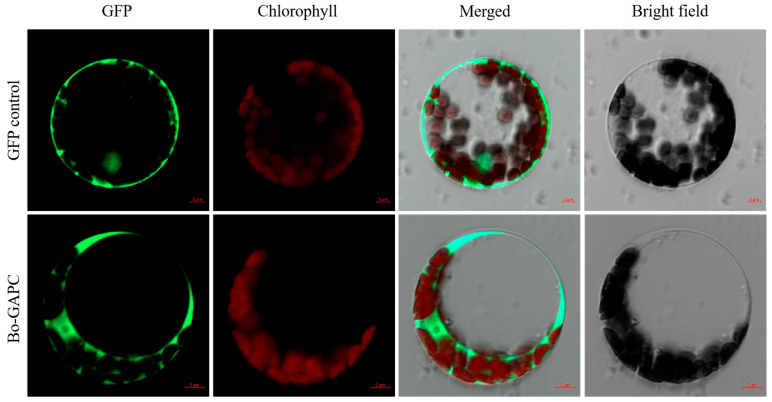
Subcellular localization of BoGAPC-GFP fusion protein. GFP control is the empty control, and Bo-GAPC is experimental. The scale is 5 μm.

**Figure 11 genes-12-01719-f011:**
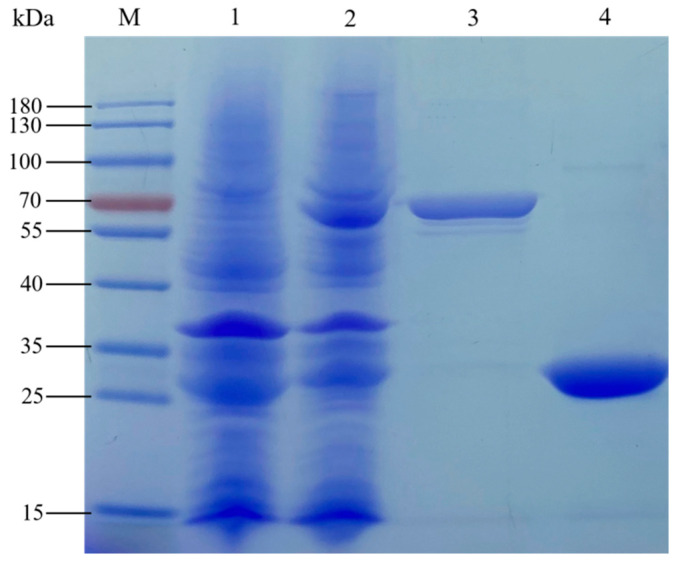
Prokaryotic expression protein analysis of BoGAPC protein. Lane 1 is uninduced pGEX-4T-1-BoGAPC protein, lane 2 is induced unpurified pGEX-4T-1-BoGAPC protein, lane 3 is induced purified pGEX-4T-1-BoGAPC protein and lane 4 is induced purified empty protein.

**Figure 12 genes-12-01719-f012:**
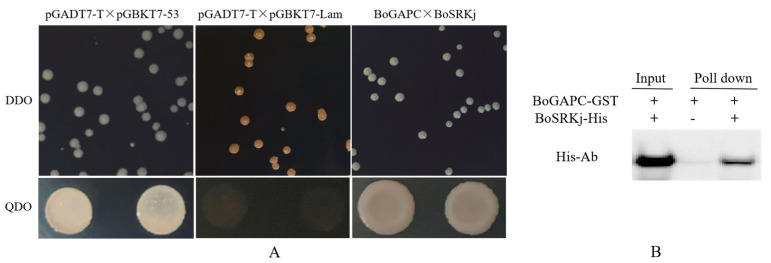
(**A**) Test protein–protein interactions on SD/-Leu/-Trp plate and SD/-Ade/-His/-Leu/-Trp plate. DDO: SD/-Leu/-Trp, QDO: SD/-Ade/-His/-Leu/-Trp. Positive control: pGADT7-T × pGBDT7-53, negative control: pGADT7-T × pGBDT7-Lam, experimental group: pGADT7-BoGAPC × pGBKT7-BoSRKj. (**B**) In vitro GST pull-down assay for the interaction of BoGAPC and BoSRKj. Purified BoSRKj-His tagged protein was pulled down by BoGAPC-GST fusion protein and was detected by Western blot analysis using His antibody.

**Table 1 genes-12-01719-t001:** *GAPDH* gene information in *B. oleracea* and *A. thaliana*.

Gene ID	Chr	Gene Length (bp)	CDS (bp)	No. of AA	PI	Molecular Weight (D)
*Bo00285s370.1*	Scaffold00285	2112	1203	400	5.03	99,916.96
*Bo02686s010.1*	Scaffold02686	1190	1026	341	5.08	82,080.15
*Bo1g152180.1*	C1	2172	1134	377	5.02	94,630.74
*Bo2g075590.1*	C2	3743	2112	703	4.92	171,516.98
*Bo2g095350.1*	C2	2666	1260	419	5.02	104,769.31
*Bo3g057250.1*	C3	2022	1017	356	5.07	83,516.28
*Bo5g016460.1*	C5	1565	1200	399	5.02	99,854.00
*Bo5g017500.1*	C5	1794	1020	339	5.06	83,749.58
*Bo5g021670.1*	C5	3001	1281	426	5.02	106,195.90
*Bo5g148380.1*	C5	2037	1017	338	5.05	83,667.04
*Bo6g080970.1*	C6	2919	1266	421	5.02	105,067.33
*Bo7g083800.1*	C7	1055	711	236	5.13	59,271.32
*Bo8g024290.1*	C8	2614	1347	448	4.99	110,675.48
*Bo8g065470.1*	C8	2061	1020	339	5.07	83,721.50
*Bo8g107420.1*	C8	2495	1071	356	5.05	88,265.82
*Bo8g107880.1*	C8	1862	1200	399	5.02	100,014.22
*AT1G12900.1*	Chr1	1950	1200	399	5.04	99,919.44
*AT1G12900.2*	Chr1	1914	954	317	5.09	79,272.45
*AT1G12900.3*	Chr1	1857	1053	350	5.08	86,837.59
*AT1G12900.4*	Chr1	1784	1053	350	5.08	86,837.59
*AT1G12900.5*	Chr1	2016	1326	441	5.01	111,216.46
*AT1G13440.1*	Chr1	2801	1017	338	5.08	83,708.07
*AT1G13440.2*	Chr1	2451	933	310	5.10	76,429.79
*AT1G16300.1*	Chr1	3365	1263	420	5.02	104,385.82
*AT1G42970.1*	Chr1	2542	1344	447	5.01	110,193.00
*AT1G79530.1*	Chr1	3506	1269	442	5.02	105,332.66
*AT3G04120.1*	Chr3	2748	1017	338	5.06	83,798.81
*AT3G26650.1*	Chr3	2183	1191	396	5.04	99,013.50

## Data Availability

This manuscript includes the essential data either as figures or as Appendix A. Publicly available datasets were analyzed in this study. This data can be found at https://www.ncbi.nlm.nih.gov/GSE107720 (accessed on 9 May 2020).

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
