# Peer review of "Analysis of SI-Related BoGAPDH Family Genes and Response of BoGAPC to SI Signal in Brassica oleracea L."

_genes, 2021, doi:10.3390/genes12111719_

Round 1

Reviewer 1 Report

Authors provided generally important results related to BoGAPC protein family analysis. Data are partially provided by available databases. Also the lack of Y2H results validation-for example in vitro by a GST-agarose pull down assay or by in vivo test -bifluorescence complementation or co-immunoprecipitation in plant material. Authors do not show or hypothesize enough the biological functions of both BoGAPC-BoSRK kinase interaction.

In my opinion Authors should at least validate the Y2H results by in vitro pull down assay (for example using GST-agarose or related material) to confirm they YH2 results by an independent test and describe better the putative biological functions of BoGAPC-BoSRK kinase interaction in discussion section.

Specific comments:

Lines: 45,47,49,66,87,94 etc- no space before brackets. Check the lack of space in other places-line 87 etc.

Section 2.4 and 2.5 Authors write of transcript structure but it is not shown in Results section.

In section 2.5 Authors performed RT-PCR not transcript structure analysis.

Line 142: Instead of fluorescent quantitative PCR should be RT-PCR or qRT-PCR. More precise description of RT-PCR experiments is necessary:

Line 143-dAction (putatively actin) as a RT-PCR standard?

How the RNA was isolated?

Assessment of RNA purity, concentration and integrity

Details of DNase treatment to remove putative remnants of genomic DNA

Reverse transcription reaction; details of reaction-temperature and time, volume and amount of used RNA

qPCR target- length of PCR product (control-actin and tested gene), target gene symbol and accession number, sequences of all RT-PCR primers. Did Authors used RT-PCR primers presented in table 1 for all 14 BoGAPC genes tested by RT-PCR? Why authors selected 14 genes from 16 BoGAPC family members?

qPCR protocol: conditions of PCR reaction, volume of reaction, concentration of magnesium ions, dNTPs, DNA polymerase type and concentration.

Reason to choose particular internal (actin) standard- citation of previous research or analysis using bestKeeper tool (Pfaffl et a. 2004)

Software and equipment used to perform and analyse RT-PCR results.

Was the qPCR performed in the same or separated tubes?

Line 162 instead of Y2Hgold should be Y2HGold

Line 167 and 168- preparing gDNA and cDNA. Amount of gDNA and cDNA as template in PCR.

Line 175,207,209  E. coli and other latin names  should be in italics

Section 2.8 no spaces before brackets

Figure 3- Authors present 10 conserved motifs- however in figure 2 are only 8 motifs.

Section 3.3

Try if detected cis-elements could be validated by co-expression test using for example Expression Angler tool.

Toufighi et. al. 2005 „The Botany Array Resource: e-Northerns, expression angling and promoter analyses”.

Expression Angler (utoronto.ca)

Section 3.5.4 If Authors have gDNA, cDNA size plus other database information they may predict the structure of transcript, or position of introns/exons. Why Authors isolated gDNA and cDNA of BoGAPC?

Section 3.5.9 and Fig. 16, it is try to say (without the standards of cell substructures) that it is cytoplasm, putatively the magnification should be larger.

Others:

Line 210-211-sentence has no sense.

Discussion 4.1 where are results of Authors? Try to show in the Discussion own results in context of work of other people.

Author Response

请参阅附件。

Reviewer 2 Report

The paper analyzed the Glyceraldehyde 3-phosphate dehydrogenase (GAPDH) genes and their response to SI signal in Brassica oleracea. The authors tried to prove that the proteins encoded by this family may be involved in the self-incompatibility reaction of B. oleracea.

Although the authors have spent a great effort to collect a large amount of data surrounding these genes, the way the paper was written and presented made it hard for readers to follow and extract meaningful results.

First, the number of figures (19 figures) is too many, make the paper too lengthy and diluted. Many can be put in the supplemental. More importantly, most figures have very poor legend descriptions, only short sentences, and tiny font sizes that are impossible to read.  These should be rewritten entirely. 

Second, it seems the authors have not reread their manuscripts, with many truncated sentences and paragraphs and grammar errors (for example, line 530…). The result section should be written in a way that makes a story and has connections between different parts.

Third, although the authors showed interactions between GAPC and SRK domains in vitro, this does not necessarily mean they are involved in SI reactions in planta. More in vivo data are needed to have a more compelling conclusion.

Round 2

Reviewer 1 Report

Authors significantly improved the manuscript. Some minor comments should be still corrected:

Comment nr 4 and 8- Authors write of dAction as reference gene – the cited paper use BoActin not Action as RT-PCR reference gene. Correct the name of reference gene in the entire text.

Comment nr 5, and 7; try combine these information and add these data for example  to a supplement file. Add to section 2.5 information that details of RNA isolation, RT reaction and qRT-PCR are presented in particular Suppl file.

Comments 9 and 10- add these information to section nr 2.5

Comment nr 12- add these information to section 2.7

Section 13- still some corrections are necessary- for example lines 149, 165, 681, 683, 688 in highlights version.

Reviewer 2 Report

The authors have spent a significant effort to improve the manuscript.